# A Survey on Medical Image Segmentation Based on Deep Learning Techniques

**Jayashree Moorthy and Usha Devi Gandhi ***

School of Information Technology and Engineering, Vellore Institute of Technology,
Vellore 632007, Tamilnadu, India
* Correspondence: ushadevi.g@vit.ac.in

**Abstract:** Deep learning techniques have rapidly become important as a preferred method for evaluating medical image segmentation. This survey analyses different contributions in the deep learning medical field, including the major common issues published in recent years, and also discusses the fundamentals of deep learning concepts applicable to medical image segmentation. The study of deep learning can be applied to image categorization, object recognition, segmentation, registration, and other tasks. First, the basic ideas of deep learning techniques, applications, and frameworks are introduced. Deep learning techniques that operate the ideal applications are briefly explained. This paper indicates that there is a previous experience with different techniques in the class of medical image segmentation. Deep learning has been designed to describe and respond to various challenges in the field of medical image analysis such as low accuracy of image classification, low segmentation resolution, and poor image enhancement. Aiming to solve these present issues and improve the evolution of medical image segmentation challenges, we provide suggestions for future research.

**Keywords:** deep learning; medical image analysis; applications; frameworks





## 1. Introduction

Deep learning approaches related to artificial intelligence (AI) algorithms, medical image segmentation, and medical image classification may have the most significant, long-term influence on a large number of individuals in a small amount of time. Convolutional neural networks are widely used in Features extraction, image formation, image recognition, and image-based representation and are all part of the technological collection and processing of medical data. Image enhancement has expanded to cover image processing methods, classification techniques, and visual extraction, including pattern classification. Deep learning techniques are used to detect accuracy and robustness and improve image enhancements.

This signified the beginning of a new area in medical imaging analysis due to considerable increases in processing capabilities, faster data storing, and multiprocessing; furthermore, it has improved dynamically ideal high-level features and semantic translation from raw data [1]. A new deep learning technique for digital data might provide human visuals that are more accurate than laboratory diagnostic predictions and could be utilized as a computer vision foundation for better healthcare decision making. In the literature, there have been many studies on using deep learning to identify infection. To our knowledge, just one survey study examining state-of-the-art research on the issue has been submitted in the last 6 years. The advancement of deep learning and its implications in medical imaging are discussed in this study [2].

Developments in the past may be used to cure a variety of illnesses, including ultrasound imaging problems, heart disease, infections, and lung damage. There is also a discussion of the number of typical deep learning access networks used in medical image analysis. However, these studies suffer from a lack of analysis of RBM, GAN, RNN,

and DNN classification techniques that exist in research developments. To discover patterns from various data types, deep learning needs the use of computer models. Deep learning is the process of identifying patterns from data collections by using computational models containing both convolutional and non-convolutional nodes of interconnected neurons [3].

A processing element is a type of cell that, like a neuron, absorbs many impulses, conducts a calculation, and then outputs a result. A feature representation is used before a linear source sequence forms in this fundamental technique. The first and most widely used deep learning technique, CNN, is highly adept at recognizing patterns in images. CNNs, similar to respiratory neural network models, are particularly associated with adaptable bases and parameters. Multiple entries are received from each neural system. Then, the data are totaled in a balanced manner. After the sum is normalized, it is input into an activation function, which gives the results.

The planning and design approach is enhanced by summarizing the existing studies and reviewing the three basic clinical methods for image segmentation and individual constraints. With the preceding challenges, a thorough analysis of the most recent deep learning-based medical image segmentation techniques is provided to support researchers in finding solutions. CNN is distinguished from human brains by the presence of fully connected layers [4].

In summary, the goal of the survey is to:

1. Identify challenges in adding deep learning techniques to the segmentation of medical imaging.
2. Identify the appropriate framework to remedy or prevent such barriers.
3. Calculate the overall effectiveness of deep learning approaches with different data sets and frameworks, as shown in Figure 1.

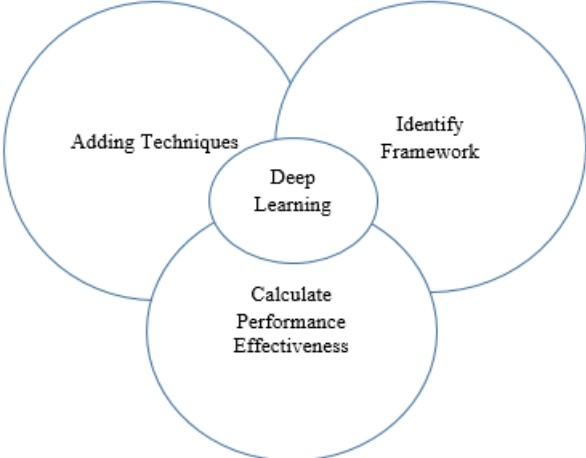

**Figure 1.** The goal of the Survey.

The remainder of the survey is divided into a number of sections. In Section 2, we present the basic outlines of deep learning approaches that have been deployed for extracting functionality for related work. In Section 3, we provide the different deep learning techniques and how they are used to contribute work in the segmentation of medical imaging. In Section 4, the importance of deep learning of essential medical image tasks is described as image classification, detection, segmentation, recording, research, and improvement. In Section 5, we review application-based profound learning frameworks. In Section 6, we calculate the overall effectiveness of performance using evaluation measures. In Section 7, we provide conclusions with an overview of performance effectiveness for improving accuracy, meaningful analysis, and a move toward future research.

## 2. Review of Related Work

Some of the deep learning techniques are added into a medical image analyses to overview the previous work to enhancement with the ideal requirements. The implementation of work is Supervised learning, weakly supervised learning methods to allow different level to identify the parameters are difficult which are used in the previous research articles [5]. A basic deep learning for medical image analysis in convolution neural network are used in this model and training process is too low [6]. The AI and DL based work is volume of enormous data is very low and process of manual, which necessitates substantial subject expertise [7]. Fully convolutional for diagnostic image classification approach used in both analytically cheap and effective and also Failed to address computational constraints [8]. Deep learning for detecting respiratory disease sensing devices and required during optimisation with Transformation compute expenses, and memory costs are also included [9]. Detecting population-related environmental risk factors size of lower labelling detail required during optimisation [10]. A methodology for extracting retinal blood vessels images using edge detection is to adaptable for segmenting retinal images and segmenting performance with a shorter run-time [11]. Deep learning and medical image segmentation is related with processing with some contribution of work, advantage and disadvantage along with different techniques are used in existing work are listed in Table 1.

**Table 1.** Literature survey in deep learning.

| Author Name | Title | Year | Contribution of Work | Advantages | Disadvantage |
|---|---|---|---|---|---|
| Chen et al. [5] | A deep learning framework for adaptive compressive sensing of high-speed train vibration responses | 2020 | Supervised learning, weakly supervised learning. | The design allowsat different levels independently. | Because numerous connections and parameters is enhanced. |
| Finck T. et al. [6] | A basic introduction to deep learning for medical image analysis | 2021 | convolutional neural networks | The networks detect the relevance of characteristics at various levels. | During the process of training is slow. |
| Kollem et al. [7] | A review of image denoising and segmentation methods based on medical images. | 2019 | AI and DL based Framework | Deep learning makes use of enormous volumes of data and time-consuming process is manual | To circumvent the time-consuming process of manual, which necessitates substantial subject expertise. |
| Bir et al. [8] | A review on medical image analysis with convolutional neural networks. | 2020 | DCNN | The dropouts approach used in both analytically cheap and effective. | Failed to address computational constraints. |
| Anwar.et al. [9] | Medical Image Analysis: A Survey of Deep Learning | 2019 | Retinal anatomy segmentation | The minimal storage size provided is often less relevant. | Requires improvement in learning process. |
| Kwekha Rashid et al. [10] | Deep learning Coronavirus disease (COVID-19) cases analysis using | 2021 | Deep Learning-based respiratory disease sensing devices. | The lower labelling detail required during optimisation. | Transformation compute expenses, and memory costs are also included. |
| O. Anwar Bég et al. [11] | A methodology for extracting retinal blood vessels from fundus images using edge detection | 2017 | This method is adaptable for segmenting retinal images and segmenting. | When compared to other techniques, it can provide superior segmentation performance with a shorter runtime. | To get a segmentation picture, a post-processing phase is performed. |

### 3. Overview of Deep Learning Techniques

Machine learning is a class of deep learning involving a large-scale neural network with several layers and features. The bulk of deep learning involves neural network architectures. Consequently, deep neural networks have become popular. Learning comprises a cascade of several levels of sophisticated diverse nodes for feature extraction and manipulation. Machine learning gives a higher level of study with relatively useful capabilities obtained from medium-access attributes, whereas lower layers learn basic characteristics close to the data input. The architecture creates a strong and modular classification model well suited to evaluating and collecting usable knowledge from massive volumes of data as well as information obtained from many sources [12].

Deep learning approaches are classified as supervised, semi-supervised, or unsupervised. In supervised learning, the network is generated by a collection of sets of containers. Every fair performance serves as a training dataset for the method of confirming the parameters of consideration. Using prior labels, the method predicts the labels of the expected output. Segmentation approaches involve a learning algorithm and may be used to recognize faces and signage, convert audio to text, and more (Figure 2).

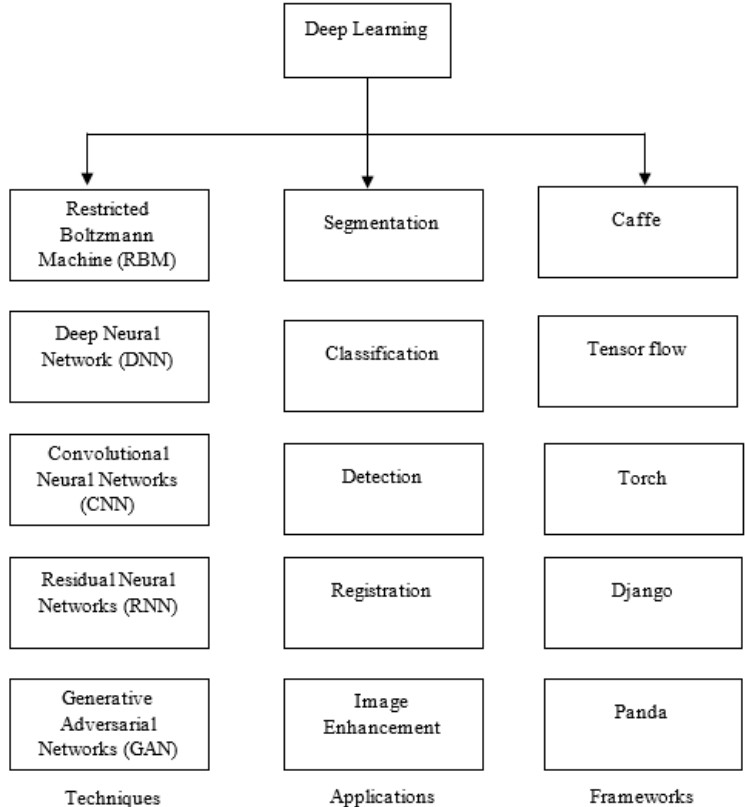

**Figure 2.** Overview of deep learning techniques, applications, and framework.

Despite the fact that convolutional neural networks have a better adaptation for medicinal image recognition, the outcomes are heavily reliant on increased labeling. In reality, many datasets with high-quality labels are uncommon, particularly in the area of medical image segmentation, due to the increased expenses of collecting and labeling data. As a result, much research based on inadequate or faulty data has been published [13].

The semi-supervised learner is a computational model that crosses the gap between supervised and unsupervised approaches. Semi-supervised validation includes both recognized and unmarked variables. Semi-supervised training is a method of learning that falls between unsupervised and supervised. Training datasets, when paired with a small number of labeled examples, can result in a large increase in correct models. Following this, some scientific implications concerning DL were drawn [14].

A restricted Boltzmann machine (RBM) is a dynamic probabilistic deep neural network that can develop a conditional probability in several sources. It became renowned after Geoffrey Hinton and associates built rapid classification techniques in the late 2000's [15].

A deep neural network is a form of neural network made up of layered limited Boltzmann machines (RBMs). RBM, a sort of randomized knn classifier, was inspired by Boltzmann machines. Despite the fact that RBM has maintained the Boltzmann machine's multiple network architecture, there is no connection between neurons at the same level, with the whole link being seen between visible and hiding layers [16].

Convolutional neural networks have long been a situation for detection algorithms, including image recognition, positioning, and separating, and are one of the most popular neural network models in use today. The significance of convolutional neural networks is due to the convolutional layers at the base of the architecture. These tiers are responsible for conducting dimension reduction and input substitution to reduce the initial issue's growth. A conventional CNN structure consists of convolutional layers and pooling layers that are fully-connected layer upon layer. For extracting the features, the convolutional layer is utilized. By reducing the amount of easy-to-train network parameters, the weight-sharing splitting feature aids in improving adaptation and avoids overfitting. Each neuron's input in this layer is coupled to a preceding layer's local receptive field. The pattern process is conducted with the pooling layer. It has the ability to minimize input dimensions while maintaining structural stability [17].

Recurrent connections are built into the RNN, allowing it to recall similarities after earlier sessions. Because the region in medical imaging is typically scattered across numerous neighboring slices, there are connections in subsequent slices. As a result, RNNs may extract multi-relations as a type of time series from the incoming layers. The RNN architecture is made up of two key sections: intra-slice data that can be collected, which may be done by any sort of CNN model, and inter-slice image retrieval, which is handled by the RNN [18].

The benefit of using the long short-term memory (LSTM) technique is that it incorporates an inter-level learning framework. Three systems are offered for sharing data to represent text: the first one uses merely an LSTM layer; the next provides an LSTM layer for each job, where each layer can utilize data from other LSTM layers; and the last one not only has the same features as the first, but also creates a bidirectional LSTM layer for all jobs. The authors in [19] show that the models help with the objectives outlined.

The ideas of generating networking and classification algorithms were used to create GANs (generative adversarial networks). This classification can tell the difference between valid and invalid data, but the proposed technique produces false intelligence. GANs are most commonly employed in applications that demand the production of pictures from text, as these networks work to improve the training program. The conceptualization layer of Google's analytical model is used to execute the concurrent nonlinear function and pooled processes in order to complete challenging tasks quickly. This is a level higher than DL that is used to automate numerous data capture operations [20].

A few of the most popular deep learning systems are the restricted Boltzmann machine (RBM), deep belief network (DBN), convolutional neural network (CNN), recurrent neural network (RNN), recursive neural network, generative adversarial networks (GAN), and direct deep reinforcement learning (DDRL) as discussed in the previous section.

## 4. Deep Learning Application for Medical Image Analysis

Medical image analysis is based on the some of the most valuable applications followed by the deep learning techniques for evaluating with medical disease like cancer, diabetics retinopathy, cardiac, lung nodule, brain tumour, fetal, thyroid, prostate. Some of the applications are related with different modals, algorithms and challenges. Here segmentation is used for process of distinguishing various portions of the image into respective classes is known as semantic segmentation. Segmentation may be considered a classification issue at the pixel level the low resolution to segment may be thought of

as a classification issue. In order to identify complex regions in medical image analysis, segmentation has been a significant study field. Image classification is one of the most researched topics in the application of deep learning in medical image analysis. At its most basic level, a classification task involves assigning three or even more identifiers to that same data from a collection of pre-determined classifications while offering an overview of classification tasks using pattern recognition. The combination of classification and positioning processes are combined in their detecting function, which involves finding the item of concern inside the picture. Deep learning for automated medical systems is an important area of research in computer-assisted diagnosis. Image enhancement is used to improve the quality of images through techniques such as demising, super-resolution, and background subtraction. As sharpness increases, processes, including classification, recognition, and fragmentation, become more appropriate and may offer a deep learning-based diagnostic visual enhancement are show in Figure 3.

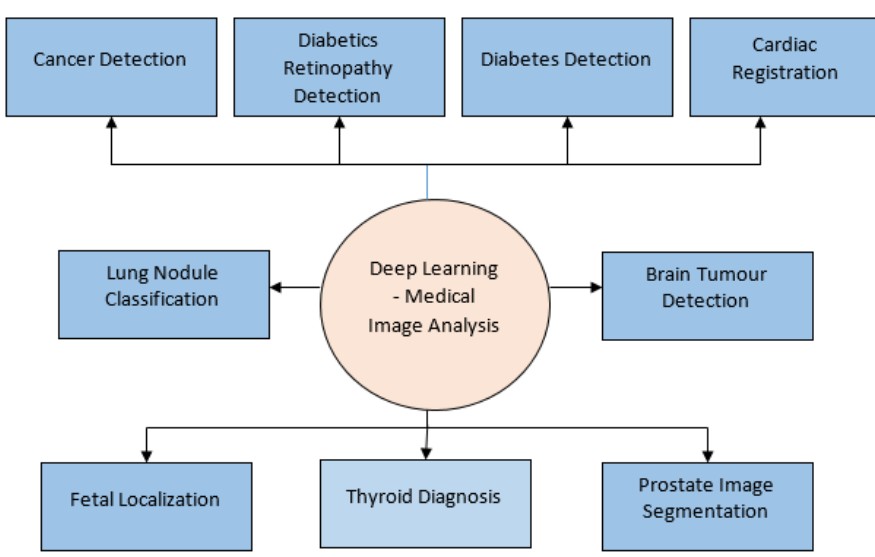

**Figure 3.** Deep learning applications in medical image analysis.

*4.1. Segmentation*

The objective of semantic segmentation is to categorize various regions of a picture into groups. At low resolution, the segment may be thought of as a classification problem. In computer-aided diagnosis, segmentation is utilized to extract anatomical sections and has been an important study field for information extraction. Here, we offer an outline of study segmentation approaches, the current state of medicinal imaging segmentation approaches, as well as the issues they confront, and possible solutions.

In recent decades, segmentation in medical images has shown impressive performance, but there is still a big issue in the shape of defective collections with classifications that are either at the pixel level or erroneous in certain locations. Currently, there is a one-of-a-kind technique to approach this problem by offering a type of training plan that utilizes a well-known normalization structure. The detection of biomarkers in medical imaging aids identification. The unsupervised DL is often used for image analysis. However, it might be erroneous because it needs a lot of exact locations and properties of both units.

The discovery of aberrant sites is the first step in the genetic sciences. MRI images using separation would provide good data regarding clinical imaging. Unfortunately, using the MRI scan segment has limitations, including a missing border region between the prostate and another organ, a multidimensional background, and a fluctuation in public characteristics. At present, the boundary-weighted domain adaptive neural network (BOWDA-Net) uses performance targets divided by cost for improving cutoff detection that is simpler in division. To overcome the problem with tiny picture data, the researchers used a boundary-weighted transfer bias algorithm, as shown in Table 2.

**Table 2.** Segmentation tasks in kidney imaging.

| References | Applications | Modals | Algorithm | Challenges |
|---|---|---|---|---|
| Denker et al. (2021) [21] | Localization | US | CNN | CNN is used with traditional characteristics to locate areas surrounding the organs. |
| Zhou et al. (2021) [22] | Segmentation | CT | ANN | 2-dimensional ANN with 32 × 32 patching, 90images validated. |
| Thong et al. (2016) [23] | Localization | CT | CNN | Combines local patch and slice based CNN. |
| Hu et al. (2016) [24] | Segmentation | CT | CNN | 3D CNN with time-implicit level levels for liver and spleen segmentation. |
| Jiang et al. (2018) [25] | Liver | tumor | CT | F-CNN U-net, coupled fCNN, and dense 3D CRF are all examples of neural networks. |
| Hoogi et al. (2017) [26] | Lesion | CT/MRI | CNN | The possibilities produced by 2D Cnn are utilized to move the segmentation method. |

*4.2. Classification*

Computer-aided diagnosis is another title for classification (CAD); in medical imaging a similar term for the categorization is diagnostic. The classifier is crucial in medical imaging evaluation. During the categorization processing phase, one or more visuals are utilized as raw data, and an individual test factor is formed as an output that characterizes the image evaluations.

Image classification is one of the most researched topics in the application of deep learning in medical image analysis. At its most basic level, a classification task involves assigning three or even more identifiers to that same data from a collection of pre-determined classifications while offering an overview of classification tasks using pattern recognition. A combination of Resnet and Imagenet dataset models beat classic CNNs in an analysis of the ImageCLEF 2017 medical imaging collection.

A suggested classifying strategy for multi-layered CNN models is based on a new weight initialization and sliding window combination. On a number of medical datasets, the system beats traditional deep learning and other intelligent systems. The non-linear and linear CNNs are trained using a wide variety of images. Depending on the specific results from a publicly available database, the proposed system performs much better than other existing systems, as shown in Table 3.

**Table 3.** Segmentation tasks in kidney imaging.

| References | Applications | Modals | Algorithm | Challenges |
|---|---|---|---|---|
| Debelee et al. (2015) [27] | Localization of prostate | MRI | ANN | Only 55 prostatitis photos were evaluated. |
| Razzak et al. (2018) [28] | Localize the fetal | UST | RBM | Some behaviour is considered that cannot be measured using typical methods. |
| Inan et al. (2022) [29] | Lung cancer detection | MRI | 3D-CNN | Tiny nodules were not detected with great precision. |
| Bai et al. (2013) [30] | Cardiac registration | MRI | DBN | Classification using many atlases Better computing performance is required. |
| Tsai et al. (2018) [31] | Cancer registration | CT | GAN | A small sample with a mild disease. |
| Maier et al. (2019) [32] | Localization | MRI | RBM | Program for 3D dynamic identification Approaches for reducing implementation time really aren't mentioned. |

*4.3. Detection*

From this, classification and positioning processes are combined in their detecting function, which involves finding the item of concern inside the picture. Deep learning for automated medical systems is an important area of research in computer-assisted diagnosis. Such analytical systems frequently need the recognition of a certain anatomical position. One of the most time-consuming jobs for specialists is identifying interesting things in categorization. Objectives usually consist of locating and identifying tiny diseases in the

whole image domain. A lengthy investigation has been conducted. Computer-aided detection techniques have a long history, meant to improve the detection accuracy or reduce the reading process of individual specialists by detecting multiple medical images. The majority of published deep learning object recognition techniques still employ CNNs for pixel (or particle) classifications, followed by some type of image acquisition to generate image suggestions.

CNN architectures and approaches are extremely similar since the classification task performed at each pixel is basically image categorization. There are several major differences between object detection and object categorization. In a learning context, the class balance is often biased heavily toward the non-object class because each pixel is categorized. Furthermore, the majority of non-object samples are rather straightforward to discriminate, preventing the deep learning approach from concentrating on the challenging ones. As a result, the challenges of implementing pattern recognition to significant item recognition are comparable to that of feature extraction. Few studies discuss image frameworks to analyze, such as class imbalance/hard-negative mining, or effective screen image augmentation. Certain fields, such as the deployment of fully convolutional multi-stream models, are expected to attract more people in the future, as shown in Table 4.

**Table 4.** Segmentation tasks in kidney imaging.

| References | Applications | Modals | Algorithm | Challenges |
|---|---|---|---|---|
| Li et al. (2015) [33] | Segmentation | CT | CNN | 3D CNN featuring moment threshold ranges for lung and organ fragmentation. |
| Wang et al. (2015) [34] | Lesion | CT | ANN | Similar strategy is repeated in two-dimensional $15 \times 15$ update categorization. |
| Hu et al. (2016) [35] | Liver | CT | CNN | Designs of Cnn using maximum entropy variable; excellent SLIVER06 outcomes. |
| Chen et al. (2020) [36] | Lesion | CT | CNN | Two-dimensional Cnn for identification of malignant tumours in follow-up CT using standard CT as feed. |

### 4.4. Registration

The technique of merging two or many images to provide more details is known as image registration. The merging of medical images improves image resolution, which aids medical practitioners in identification. Because the medical image provided by a technique such as CNN may not give all of the data requested by medical experts, it is important to combine data with additional methods. When the input data are combined, for example, more information is obtained than if the data were obtained individually. Many studies employing deep learning techniques for image registration have been conducted. A 3D CNN-based architecture for imagery identification of encouragement representations was proposed. The given input data sets were used to evaluate the suggested technique's efficiency. For unsupervised flexible image registration, they presented a hierarchical converters technique from beginning to conclusion. This proposed technology was validated in complex 3D biomedical screening, where deformation registration is most commonly used, and showed persistent and considerable improved performance over other current approaches.

1. To implement a continuously proposed methodology, a profound trained model is often used to produce predictors of success between diverse images;
2. Using deep prediction systems to directly predict transformation parameters. The suggested strategy outperforms state-of-the-art techniques with minimal variables and quick registration durations according to experimental observations.

The suggested approach uses slice iteration to improve registration efficiency between images with large and sudden evolutions. The suggested approach was tested on a genuine dataset and found to be satisfactory.

### *4.5. Image Enhancement*

Image enhancement is used to improve the quality of images through techniques such as demising, super-resolution, and background subtraction. As sharpness increases, processes, including classification, recognition, and fragmentation, become more appropriate and may offer a deep learning-based diagnostic visual enhancement solution. Deep learning-based demising approaches rely on the usage of data preparation. The deep connected generating model developed using fully connected layers performed well with limited data. Even in elevated situations, their approach was able to retrieve data. To overcome the limitations posed by short datasets, one should adopt a residual learning strategy. The scans are sent to CNN for classification, and the information on the files is publicly disclosed. During image classification, every new stage in diagnosis is visually localized, which adds a region of interest around the appropriate area—a process known as classifying with tracking. The word localization refers to determining the illness in the picture. The pre-processing step of a medical assessment in which the radiologist recognizes some critical traits is called anatomical localization. DL simulations are often used to localize the illness in various research studies in recent years. The investigators, for example, provided a deep CNN-based model for organ or body component categorization.

## 5. Deep Learning Frameworks

The previous research work may utilize a variety of deep learning frameworks to accomplish deep learning approaches. The following are the many deep learning frameworks and their versions:

1. Caffe is a deep learning framework for CNN models that is built on MKL, OpenBLAS, and cuBLAS, among other computer packages. Caffe comes with an application of techniques for learning, forecasting, delicate, and so forth. It also includes a number of modeling purposes and procedures for students to use. Caffe's application servers are simple to install. It also comes with Matlab and Python interfaces that are easy to use. Caffe is simpler to study than other frameworks, which is why many novices prefer it.
2. TensorFlow is a large-scale platform learning framework that allows machine learning algorithms to be executed through an application. Voice control, data analysis, nanotechnology, knowledge representation, and computational linguistics all seem to be examples of advanced technologies. These are just a few of the applications and off shoots of the making technique to transfer calculations specified using and involving the handling concept onto several device types, including Android and Apple. It also allows processing many devices on a handheld product over several networks.
3. Torch is capable of supporting the majority of machine learning methods. Multi-layer perceptions, additional frequent approaches, and concepts include SVM classifiers, maximum entropy processes, Markov models, space-time convolutional neural networks, logistic regression, probabilistic classifications, and others. The torch may be integrated into Apple, Android, and FPGA in addition to supporting CPU and GPU.
4. Keras is a high access soft computing framework built on Programming language. The concept was developed by Francois Chollet, a Microsoft AI-powered developer. Microsoft, Visa, Youtube, Cisco, and Taxi, among several others, are now using Keras. This lecture covers Keras implementations, supervised learning bases, Keras modeling, Keras layering, Keras components, and finally numerous significant implementations.
5. Django is an elevated Programming language development tool for easily creating secure and functioning websites. Django is a programming environment created by software practitioners that does the hard lifting so you can work on designing your program rather than spinning the wheel. It is freely available, with a thriving community, huge backing, and a range of free model support options.

## 6. Performance Efficiency

From Deep Learning, overall performance efficiency is calculated using evaluation metrics to find the accuracy, sensitivity, specificity, and overall accuracy and loss based on the techniques and frameworks in Table 5.

**Table 5.** Overall performance efficiency.

| Author name | Datasets | Algorithm | Efficiency |
|---|---|---|---|
| Ozturk et al. (2017) [37] | OLS ARIMA | RBM | Accuracy 78.6, Sensitivity 75.09, Specificity 67.4. |
| Kumar, et al. (2021) [38] | Chest X-ray | DBN | Accuracy 87.6, Sensitivity 84.2, Specificity 67.4. |
| Butt et al. (2019) [39] | Electronic medical records | DNN | Accuracy 90.24, Sensitivity 91.02, Specificity 80.55. |
| Budd et al. (2021) [40] | CT scan Images | CNN | Accuracy 98.08, Sensitivity 98.02, Specificity 92.4. |
| Paarth Bir et al. (2019) [41] | MRI scan images | RNN | Accuracy 97.08. |
| Lekadir et al. (2021) [42] | UST scan images | LSTM | Test Accuracy 90.3. |
| Xun, et al. (2022) [43] | CT exams | GAN | Accuracy 89.02, Specificity 91.2. |

Based on the previous survey of performance efficiency, we present a graphical representation of the work from 2017 to 2021 using deep learning techniques and deep leaning applications with different deep learning frameworks (Figures 4–6).

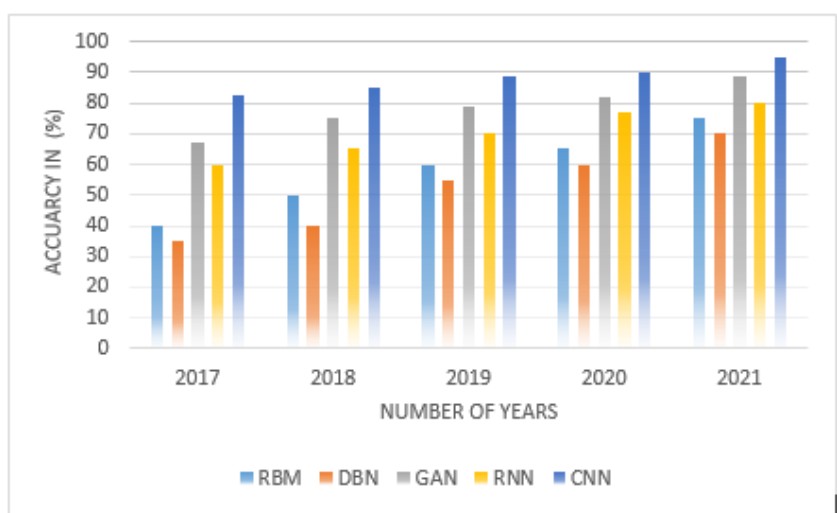

**Figure 4.** Performance efficiency with different deep learning techniques.

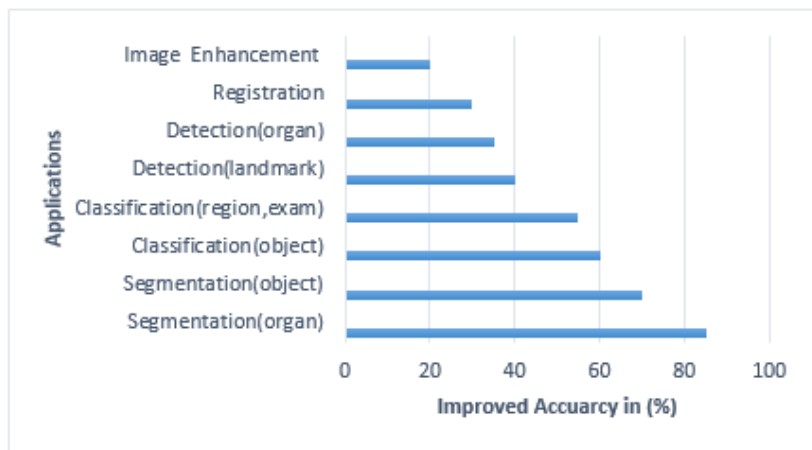

**Figure 5.** Accuracy vs. deep learning applications.

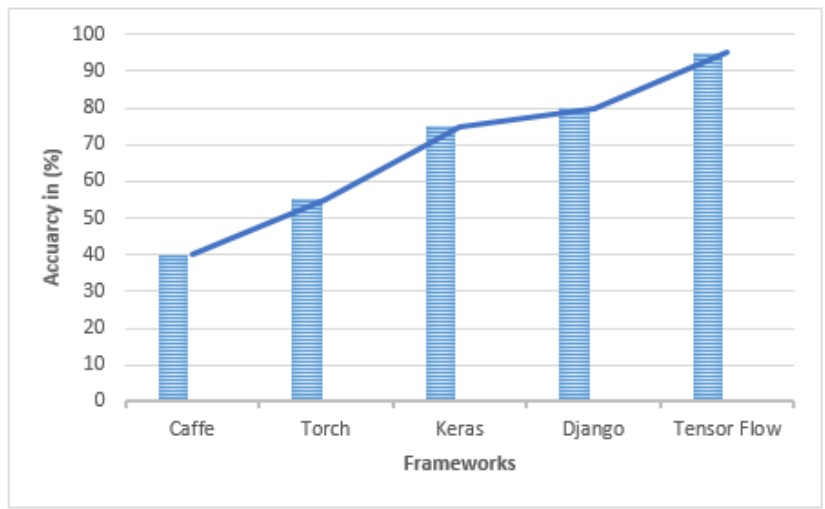

**Figure 6.** Accuracy vs. deep learning frameworks.

## 7. Conclusions

Deep learning has made progress in medical image analysis to increase in a significant way, but the effect of image classification, segmentation, and registration still limits its practical application. It has shown promising results as processing abilities, and the availability of structured data for use has improved. However, there are a number of barriers that must be overcome even before automatic processes can be implemented. It is possible to use new technologies that are designed for a number of roles, and various frameworks are available for building profound learning systems, which were explored here. Deep learning has been designed to describe and respond to various challenges in the field of medical image analysis, including low accuracy of image classification, low segmentation resolution, and poor image enchantment.

From the evaluation, deep learning techniques can analyze various amounts of information to create deep learning applications and frameworks that would contribute to the success and better classification accuracy, increasing the resolution with segmentation, and improving image enhancement by using convolutional neural networks. Since deep learning technology is still in its early stages, it is still possible to develop new models with better accuracy. Apart from this, improvements can be made in the diagnostic and modeling sub-domains. In order for convolutional neural networks to be used in the identification of medical diseases, it is essential to ensure that future research continues on track, ultimately improving the effectiveness of disease detection techniques. Other research work could be used in the description provided to arrange their research contributions and activities.

The scale of the current medical image data sets is small. Deep learning algorithms need a lot of data set support to be trained, which causes overfitting issues when deep learning models are trained. The recommended future research could be improved by expanding feature extraction with high-resolution data for better image segmentation, classification, and enhancements with different advanced methods, applications, and frameworks for medical challenge-aided deep learning.

**Author Contributions:** Conceptualization, supervision, methodology, visualization, software, validation, formal analysis, investigation, resources, writing—review and editing, J.M.; conceptualization, methodology, and writing—review and editing, U.D.G. All authors have read and agreed to the published version of the manuscript.

**Funding:** This research received no external funding.

**Institutional Review Board Statement:** Not Applicable.

**Informed Consent Statement:** Not Applicable.

**Data Availability Statement:** Not Applicable.

**Conflicts of Interest:** The authors declare no conflict of interest.

## Abbreviations

The following abbreviations are used in this manuscript:

| | |
|---|---|
| CNN | Convolutional Neural Networks |
| AI | Artificial Intelligence |
| DL | Deep Learning |
| RBM | Restricted Boltzmann Machine |
| DNN | Deep Neural Networks |
| RNN | Residual Neural Networks |
| GAN | Generative Adversarial Networks |
| LSTM | Long Short-Term memory |
| DBN | Deep Belief Network |
| DDRL | Direct Deep Reinforcement Learning |
| UST | Therapeutic Ultrasound |
| CT | Computed tomography |
| ANN | Artificial Neural Networks |
| MRI | Magnetic Resonance Imaging |
| MKL | Math Kernel Library |

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
