# Peer review of "A Survey on Medical Image Segmentation Based on Deep Learning Techniques"

_2504-2289, doi:10.3390/bdcc6040117_

Round 1

Reviewer 1 Report

The suggested paper is related with using deep-learning techniques in the field of medical image analysis research, and suggests the research trends and the direction for meaningful future studies. I think it can be helpful to readers about what the paper claims and presents, but it would be nice to present more clear and meaningful suggestions with the wider cases. It would be nice to suggest clearer conclusions and suggestions on future research topics.

Reviewer 2 Report

In this paper, the authors survey deep learning research in medical image analysis.

1. We compare the description of the abstract with the conclusion description after the article and feel that abstract description cannot highlight the focus of the article. It is recommended to rewrite the main summary of the abstract describing the conclusion.

2. Although these papers are worth studying. But authors must highlight their contributions and usefulness.

3. The literature review of this paper is poor, and the quality of this paper needs to be improved.

4. I cannot find anything novel in this paper. The authors need to review related work on how the new technology is addressed.

5. In general, the article needs to mention whether the goal was created or how to move towards effective implementation. 

Reviewer 3 Report

Dear authors,

I am happy to review this paper.  

This paper presents a survey of deep learning techniques in medical imaging. After, a careful review of this paper I have decided to reject this paper. Please look at the following comments and consider them for the future submission of this work.

1).  There are grammatical errors in the abstract that affects the readability of this work. For example line 6 (Based on the study, in the following areas the way.), lines 10 -11 (Finally, a summary of the present state of the art, a critical evaluation of outstanding challenges, and a conclusion with research suggestions for future research.) are not complete sentences.

2).  No space between two sentences on line 25.

3).  On line 31, the work you cited discussed medical image classification only and not medical imaging in general. This is not consistent with your citation.

4). The definition of deep learning on lines 37-39 is not correct because deep learning also extends to deep non-convolutional networks.  Also, I don't think the cited paper on line 39 discussed anything about deep learning. 

 5).  Lines 42 - 44 misrepresent deep learning. CNN is not an algorithm it is a type of deep network architecture. 

6). There is a typographical error on line 78 ("trainin" should be "training")

7) Figure 2 shows the overview of deep learning techniques. RBM, CNN, RNN, and GAN are types of DNN. This is not clear from the figure.

 For future submission, reorganize the content and also keep definitions of the terms such as deep learning and CNN  consistent with the existing literature in this field.

Thanks

Round 2

Reviewer 1 Report

Authors made the paper more clear and valuable than first-version.

Anyway, to make it more valuable, I recommend that authors have to do double-check for spells, grammar, and style of the paper.

Reviewer 2 Report

The authors have improved the quality of this research manuscript by adding recent enhancements to deep learning and medical image segmentation by making changes based on various shortcomings of this manuscript.

The themes of this manuscript and the application of technological developments are worthy of study to contribute to the development of industrial technologies.

Reviewer 3 Report

Dear Author(s),

I have gone through the changes you listed in your response. I think the paper can be accepted.

Thanks.
